# ASK THE RIGHT QUESTIONS:
# ACTIVE QUESTION REFORMULATION WITH REINFORCEMENT LEARNING

**Christian Buck, Jannis Bulian, Massimiliano Ciaramita, Wojciech Gajewski,
Andrea Gesmundo, Neil Houlsby, Wei Wang**
Google
`{cbuck,jbulian,massi,wgaj,agesmundo,neilhoulsby,wangwe}@google.com`

## ABSTRACT

We frame Question Answering (QA) as a Reinforcement Learning task, an approach that we call *Active Question Answering*. We propose an agent that sits between the user and a black box QA system and learns to reformulate questions to elicit the best possible answers. The agent probes the system with, potentially many, natural language reformulations of an initial question and aggregates the returned evidence to yield the best answer. The reformulation system is trained end-to-end to maximize answer quality using policy gradient. We evaluate on SearchQA, a dataset of complex questions extracted from *Jeopardy!*. The agent outperforms a state-of-the-art base model, playing the role of the environment, and other benchmarks. We also analyze the language that the agent has learned while interacting with the question answering system. We find that successful question reformulations look quite different from natural language paraphrases. The agent is able to discover non-trivial reformulation strategies that resemble classic information retrieval techniques such as term re-weighting (tf-idf) and stemming.

## 1   INTRODUCTION

Web and social media have become primary sources of information. Users' expectations and information seeking activities co-evolve with the increasing sophistication of these resources. Beyond navigation, document retrieval, and simple factual question answering, users seek direct answers to complex and compositional questions. Such search sessions may require multiple iterations, critical assessment, and synthesis (Marchionini, 2006).

The productivity of natural language yields a myriad of ways to formulate a question (Chomsky, 1965). In the face of complex information needs, humans overcome uncertainty by reformulating questions, issuing multiple searches, and aggregating responses. Inspired by humans' ability to *ask the right questions*, we present an agent that learns to carry out this process for the user. The agent sits between the user and a backend QA system that we refer to as 'the environment'. We call the agent AQA, as it implements an *active question answering* strategy. AQA aims to maximize the chance of getting the correct answer by sending a reformulated question to the environment. The agent seeks to find the best answer by asking many questions and aggregating the returned evidence. The internals of the environment are not available to the agent, so it must learn to probe a black-box optimally using only question strings. The key component of the AQA agent is a sequence-to-sequence model trained with reinforcement learning (RL) using a reward based on the answer returned by the environment. The second component to AQA combines the evidence from interacting with the environment using a convolutional neural network to select an answer.

We evaluate on a dataset of *Jeopardy!* questions, SearchQA (Dunn et al., 2017). These questions are hard to answer by design because they use convoluted language, e.g., *Travel doesn't seem to be an issue for this sorcerer & onetime surgeon; astral projection & teleportation are no prob* (answer: *Doctor Strange*). Thus SearchQA tests the ability of AQA to reformulate questions such that the QA system has the best chance of returning the correct answer. AQA improves over the performance of a deep network built for QA, BiDAF (Seo et al., 2017a), which has produced state-of-the-art

results on multiple tasks, by 11.4% absolute F1, a 32% relative F1 improvement. Additionally, AQA outperforms other competitive heuristic query reformulation benchmarks.

AQA defines an instance of machine-machine communication. One side of the conversation, the AQA agent, is trying to adapt its language to improve the response from the other side, the QA environment. To shed some light on this process we perform a qualitative analysis of the language generated by the AQA agent. By evaluating on MSCOCO (Lin et al., 2014), we find that the agent's question reformulations diverge significantly from natural language paraphrases. Remarkably, though, the agent is able to learn non-trivial and transparent policies. In particular, the agent is able to discover classic IR query operations such as term re-weighting, resembling tf-idf, and morphological simplification/stemming. A possible reason being that current machine comprehension tasks involve the ranking of short textual snippets, thus incentivizing relevance, more than deep language understanding.

## 2 RELATED WORK

Lin & Pantel (2001) learned patterns of question variants by comparing dependency parsing trees. Duboue & Chu-Carroll (2006) showed that MT-based paraphrases can be useful in principle by providing significant headroom in oracle-based estimations of QA performance. Recently, Berant & Liang (2014) used paraphrasing to augment the training of a semantic parser by expanding through the paraphrases as a latent representation. Bilingual corpora and MT have been used to generate paraphrases by pivoting through a second language. Recent work uses neural translation models and multiple pivots (Mallinson et al., 2017). In contrast, our approach does not use pivoting and is, to our knowledge, the first direct neural paraphrasing system. Riezler et al. (2007) propose phrase-based paraphrasing for query expansion. In contrast with this line of work, our goal is to generate full question reformulations while optimizing directly the end-to-end target performance metrics.

Reinforcement learning is gaining traction in natural language understanding across many problems. For example, Narasimhan et al. (2015) use RL to learn control policies for multi-user dungeon games where the state of the game is summarized by a textual description, and Li et al. (2016) use RL for dialogue generation. Policy gradient methods have been investigated recently for MT and other sequence-to-sequence problems. They alleviate limitations inherent to the word-level optimization of the cross-entropy loss, allowing the use of sequence-level reward functions, like BLEU. Reward functions based on language models and reconstruction errors are used to bootstrap MT with fewer resources (Xia et al., 2016). RL training can also prevent *exposure bias*; an inconsistency between training and inference time stemming from the fact that the model never sees its own mistakes during training (Ranzato et al., 2015). We also use policy gradient to optimize our agent, however, we use end-to-end question answering quality as the reward.

Uses of policy gradient for QA include Liang et al. (2017), who train a semantic parser to query a knowledge base, and Seo et al. (2017b) who propose query reduction networks that transform a query to answer questions that involve multi-hop common sense reasoning. The work of Nogueira & Cho (2016) is most related to ours. They identify a document containing an answer to a question by following links on a graph. Evaluating on a set of questions from the game *Jeopardy!*, they learn to walk the Wikipedia graph until they reach the predicted answer. In a follow-up, Nogueira & Cho (2017) improve document retrieval with an approach inspired by relevance feedback in combination with RL. They reformulate a query by adding terms from documents retrieved from a search engine for the original query. Our work differs in that we generate complete sequence reformulations rather than adding single terms, and we target question-answering rather than document retrieval.

Active QA is also related to recent research on fact-checking: Wu et al. (2017) propose to perturb database queries in order to estimate the support of quantitative claims. In Active QA questions are perturbed semantically with a similar purpose, although directly at the surface natural language form.

## 3 ACTIVE QUESTION ANSWERING MODEL

Figure 1 shows the Active Question Answering (AQA) agent-environment setup. The AQA model interacts with a black-box environment. AQA queries it with many versions of a question, and finally returns the best of the answers found. An episode starts with an original question $q_0$. The agent then

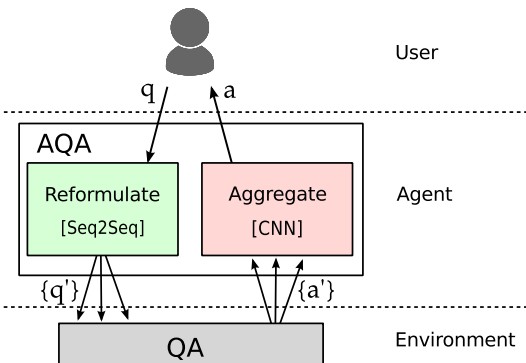

Figure 1: The AQA agent-environment setup. In the downward pass the agent reformulates the question and sends variants to the QA system. In the upward pass the final answer is selected.

generates a set of reformulations $\{q_i\}_{i=1}^N$. These are sent to the environment which returns answers $\{a_i\}_{i=1}^N$. The selection model then picks the best from these candidates.

### 3.1 QUESTION-ANSWERING ENVIRONMENT

For the QA environment, we use a competitive neural question answering model, BiDirectional Attention Flow (BiDAF) (Seo et al., 2017a). BiDAF is an extractive QA system, it selects answers from contiguous spans of a given document. Given a question, the environment returns an answer and, during training, a reward. The reward may be any quality metric for the returned answer, we use token-level F1 score. Note that the reward for each answer $a_i$ is computed against the original question $q_0$. We assume that the environment is opaque; the agent has no access to its parameters, activations or gradients. This setting enables one, in principle, to also interact with other information sources, possibly providing feedback in different modes such as images and structured data from knowledge bases. However, without propagating gradients through the environment we lose information, feedback on the quality of the question reformulations is noisy, presenting a challenge for training.

### 3.2 REFORMULATION MODEL

The reformulator is a sequence-to-sequence model, as is popular for neural machine translation. We build upon the implementation of Britz et al. (2017). The major departure from the standard MT setting is that our model reformulates utterances in the same language. Unlike in MT, there is little high quality training data available for monolingual paraphrasing. Effective training of highly parametrized neural networks relies on an abundance of data. We address this challenge by first pre-training on a related task, multilingual translation, and then using signals produced during the interaction with the environment for adaptation.

### 3.3 ANSWER SELECTION MODEL

During training, we have access to the reward for the answer returned for each reformulation $q_i$. However, at test time we must predict the best answer $a^*$. The selection model selects the best answer from the set $\{a_i\}_{i=1}^N$ observed during the interaction by predicting the difference of the F1 score to the average F1 of all variants. We use pre-trained embeddings for the tokens of query, rewrite, and answer. For each, we add a 1-dimensional CNN followed by max-pooling. The three resulting vectors are then concatenated and passed through a feed-forward network which produces the output.

## 4 TRAINING

### 4.1 QUESTION ANSWERING ENVIRONMENT

We train a model on the training set for the QA task at hand, see Section 5.4 for details. Afterwards, BiDAF becomes the black-box environment and its parameters are not updated further. In principle,

we could train both the agent and the environment jointly to further improve performance. However, this is not our desired task: our aim is for the agent to learn to communicate using natural language with an environment over which is has no control.

## 4.2 Policy Gradient Training of the Reformulation Model

For a given question $q_0$, we want to return the best possible answer $a^*$, maximizing a reward $a^* = \mathrm{argmax}_a R(a|q_0)$. Typically, $R$ is the token level F1 score on the answer. The answer $a = f(q)$ is an unknown function of a question $q$, computed by the environment. The reward is computed with respect to the original question $q_0$ while the answer is provided for $q$. The question is generated according to a policy $\pi_\theta$ where $\theta$ are the policy's parameters $q \sim \pi_\theta(\,\cdot\,|q_0)$. The policy, in this case, a sequence-to-sequence model, assigns a probability

$$\pi_\theta(q|q_0) = \prod_{t=1}^{T} p(w_t|w_1, \ldots, w_{t-1}, q_0) \tag{1}$$

to any possible question $q = w_1, \ldots, w_T$, where $T$ is the length of $q$ with tokens $w_t \in V$ from a fixed vocabulary $V$. The goal is to maximize the expected reward of the answer returned under the policy, $\mathbb{E}_{q \sim \pi_\theta(\cdot|q_0)}[R(f(q))]$. We optimize the reward directly with respect to parameters of the policy using Policy Gradient methods (Sutton & Barto, 1998). The expected reward cannot be computed in closed form, so we compute an unbiased estimate with Monte Carlo sampling,

$$\mathbb{E}_{q \sim \pi_\theta(\cdot|q_0)}[R(f(q))] \approx \frac{1}{N} \sum_{i=1}^{N} R(f(q_i)), \quad q_i \sim \pi_\theta(\,\cdot\,|q_0) \tag{2}$$

To compute gradients for training we use REINFORCE (Williams & Peng, 1991),

$$\nabla \mathbb{E}_{q \sim \pi_\theta(\cdot|q_0)}[R(f(q))] = \mathbb{E}_{q \sim \pi_\theta(\cdot|q_0)} \nabla_\theta \log(\pi_\theta(q|q_0))R(f(q)) \tag{3}$$

$$\approx \frac{1}{N} \sum_{i=1}^{N} \nabla_\theta \log(\pi(q_i|q_0))R(f(q_i)), \quad q_i \sim \pi_\theta(\,\cdot\,|q_0) \tag{4}$$

This estimator is often found to have high variance, leading to unstable training (Greensmith et al., 2004). We reduce the variance by subtracting the following baseline reward: $B(q_0) = \mathbb{E}_{q \sim \pi_\theta(\cdot|q_0)}[R(f(q))]$. This expectation is also computed by sampling from the policy given $q_0$.

We often observed collapse onto a sub-optimal deterministic policy. To address this we use entropy regularization

$$H[\pi_\theta(q|q_0)] = -\sum_{t=1}^{T} \sum_{w_t \in V} p_\theta(w_t|w_{<t}, q_0) \log p_\theta(w_t|w_{<t}, q_0) \tag{5}$$

This final objective is:

$$\mathbb{E}_{q \sim \pi_\theta(\cdot|q_0)}[R(f(q)) - B(q_0)] + \lambda H[\pi(q|q_0)], \tag{6}$$

where $\lambda$ is the regularization weight.

## 4.3 Answer Selection

Unlike the reformulation policy, we train the answer with either beam search or sampling. We can produce many rewrites of a single question from our reformulation system. We issue each rewrite to the QA environment, yielding a set of (query, rewrite, answer) tuples from which we need to pick the best instance. We train another neural network to pick the best answer from the candidates. We frame the task as binary classification, distinguishing between above and below average performance. In training, we compute the F1 score of the answer for every instance. If the rewrite produces an answer with an F1 score greater than the average score of the other rewrites the instance is assigned a positive label. We ignore questions where all rewrites yield equally good/bad answers. We evaluated FFNNs, LSTMs, and CNNs and found that the performance of all systems was comparable. We choose a CNN which offers good computational efficiency and accuracy (cf. 3.3).

### 4.4 PRETRAINING OF THE REFORMULATION MODEL

We pre-train the policy by building a paraphrasing Neural MT model that can translate from English to English. While parallel corpora are available for many language pairs, English-English corpora are scarce. We first produce a multilingual translation system that translates between several languages (Johnson et al., 2016). This allows us to use available bilingual corpora. Multilingual training requires nothing more than adding two special tokens to every line which indicate the source and target languages. The encoder-decoder architecture of the translation model remains unchanged.

As Johnson et al. (2016) show, this model can be used for *zero-shot translation*, i.e. to translate between language pairs for which it has seen no training examples. For example, after training English-Spanish, English-French, French-English, and Spanish-English the model has learned a single encoder that encodes English, Spanish, and French and a decoder for the same three languages. Thus, we can use the same model for French-Spanish, Spanish-French and also English-English translation by adding the respective tokens to the source. Johnson et al. (2016) note that zero-shot translation usually performs worse than bridging, an approach that uses the model twice: first, to translate into a pivot language, then into the target language. However, the performance gap can be closed by running a few training steps for the desired language pair. Thus, we first train on multilingual data, then on a small corpus of monolingual data.

## 5 EXPERIMENTS

### 5.1 QUESTION ANSWERING DATA AND BIDAF TRAINING

SearchQA (Dunn et al., 2017) is a dataset built starting from a set of *Jeopardy!* clues. Clues are obfuscated queries such as *This 'Father of Our Country' didn't really chop down a cherry tree*. Each clue is associated with the correct answer, e.g. *George Washington*, and a list of snippets from Google's top search results. SearchQA contains over 140k question/answer pairs and 6.9M snippets. We train our model on the pre-defined training split, perform model selection and tuning on the validation split and report results on the validation and test splits. The training, validation and test sets contain 99,820, 13,393 and 27,248 examples, respectively.

We train BiDAF directly on the SearchQA training data. We join snippets to form the context from which BiDAF selects answer spans. For performance reasons, we limit the context to the top 10 snippets. This corresponds to finding the answer on the first page of Google results. The results are only mildly affected by this limitation, for 10% of the questions, there is no answer in this shorter context. These data points are all counted as losses. We trained with the Adam optimizer for 4500 steps, using learning rate 0.001, batch size 60.

### 5.2 QUESTION REFORMULATOR TRAINING

For the pre-training of the reformulator, we use the multilingual United Nations Parallel Corpus v1.0 (Ziemski et al., 2016). This dataset contains 11.4M sentences which are fully aligned across six UN languages: Arabic, English, Spanish, French, Russian, and Chinese. From all bilingual pairs, we produce a multilingual training corpus of 30 language pairs. This yields 340M training examples which we use to train the zero-shot neural MT system (Johnson et al., 2016). We tokenize our data using 16k sentence pieces.[1] Following Britz et al. (2017) we use a bidirectional LSTM as the encoder and a 4-layer stacked LSTM with attention as the decoder. The model converged after training on 400M instances using the Adam optimizer with a learning rate of 0.001 and batch size of 128.

The model trained as described above has poor quality. For example, for the question *What month, day and year did Super Bowl 50 take place?*, the top rewrite is *What month and year goes back to the morning and year?*. To improve quality, we resume training on a smaller monolingual dataset, extracted from the Paralex database of question paraphrases (Fader et al., 2013).[2] Unfortunately, this data contains many noisy pairs. We filter many of these pairs out by keeping only those where the Jaccard coefficient between the sets of source and target terms is above 0.5. Further, since the number of paraphrases for each question can vary significantly, we keep at most 4 paraphrases for each question. After processing, we are left with about 1.5M pairs out of the original 35M. The

---

[1] https://github.com/google/sentencepiece
[2] http://knowitall.cs.washington.edu/paralex/

refined model has visibly better quality than the zero-shot one; for the example question above it generates *What year did superbowl take place?*. We also tried training on the monolingual pairs alone. As in (Johnson et al., 2016), the quality was in between the multilingual and refined models.

After pre-training the reformulator, we switch the optimizer from Adam to SGD and train for 100k RL steps of batch size 64 with a low learning rate of 0.001. We use an entropy regularization weight of $\lambda = 0.001$. For a stopping criterion, we monitor the reward from the best single rewrite, generated via greedy decoding, on the validation set. In contrast to our initial training which we ran on GPUs, this training phase is dominated by the latency of the QA system and we run inference and updates on CPU and the BiDAF environment on GPU.

## 5.3 Training the Answer Selector

For the selection model we use supervised learning: first, we train the reformulator, then we generate $N = 20$ rewrites for each question in the SearchQA training and validation sets. After sending these to the environment we have about 2M (question, rewrite, answer) triples. We remove queries where all rewrites yield identical rewards, which removes about half of the training data. We use pre-trained 100-dimensional embeddings (Pennington et al., 2014) for the tokens. Our CNN-based selection model encodes the three strings into 100-dimensional vectors using a 1D CNN with kernel width 3 and output dimension 100 over the embedded tokens, followed by max-pooling. The vectors are then concatenated and passed through a feed-forward network which produces the binary output, indicating whether the triple performs below or above average, relative to the other reformulations and respective answers.

We use the training portion of the SearchQA data thrice, first for the initial training of the BiDAF model, then for the reinforcement-learning based tuning of the reformulator, and finally for the training of the selector. We carefully monitored that this didn't cause severe overfitting. BiDAF alone has a generalization gap between the training and validation set errors of 3.4 F1. This gap remains virtually identical after training the rewriter. After training the CNN, AQA-Full has a slightly larger gap of 3.9 F1. We conclude that training AQA on BiDAF's training set causes very little additional overfitting. We use the test set only for evaluation of the final model.

## 5.4 Baselines and Benchmarks

As a baseline, we report the results of the modified pointer network, called Attention Sum Reader (ASR), developed for SearchQA (Dunn et al., 2017).We also report the performance of the BiDAF environment used without the reformulator to answer the original question.

We evaluate against several benchmarks. First, following Kumaran & Carvalho (2009), we implement a system (MI-SubQuery) that generates reformulation candidates by enumerating all subqueries of the original SearchQA query and then keeps the top $N$ ranked by mutual information.[3] From this set, we pick the highest scoring one as the top hypothesis to be used as a single rewrite. We also use the whole set to train a CNN answer selector for this specific source of rewrites. In this way, we can compare systems fairly both in single prediction or ensemble prediction modes. Additionally, we evaluate against another source of reformulations: the zero-shot monolingual NMT system trained on the U.N. corpus and Paralex (Base-NMT), without reinforcement learning. As with the MI-SubQuery benchmark, we evaluate the Base-NMT system both as a single reformulation predictor and as a source of $N$ best rewrites, for which we train a dedicated CNN answer selector. We also report human performance on SearchQA, based on a sample of the test set, from (Dunn et al., 2017).

## 5.5 Results

We evaluate several variants of AQA. For each query $q$ in the evaluation we generate a list of reformulations $q_i$, for $i = 1 \ldots N$, from the AQA reformulator trained as described in Section 4. We set $N = 20$ in these experiments, the same value is used for the benchmarks. In *AQA TopHyp* we use the top hypothesis generated by the sequence model, $q_1$. In *AQA Voting* we use BiDAF scores

---

[3]As in (Kumaran & Carvalho, 2009), we choose the reformulations from the term-level subsequences of length 3 to 6. We associate each reformulation with a graph, where the vertices are the terms and the edges are the mutual information between terms with respect to the collection of contexts. We rank the reformulations by the average edge weights of Maximum Spanning Trees of the corresponding graphs.

| | | Baseline | | MI-SubQuery | | Base-NMT | | AQA | | | | |
|---|---|---|---|---|---|---|---|---|---|---|---|---|
| | | ASR | BiDAF | TopHyp | CNN | TopHyp | CNN | TopHyp | Voting | MaxConf | CNN | Human |
| Dev | EM | - | 31.7 | 24.1 | 37.5 | 26.0 | 37.5 | 32.0 | 33.6 | 35.5 | **40.5** | - |
| | F1 | 24.2 | 37.9 | 29.9 | 44.5 | 32.2 | 44.8 | 38.2 | 40.5 | 42.0 | **47.4** | - |
| Test | EM | - | 28.6 | 23.2 | 35.8 | 24.8 | 35.7 | 30.6 | 33.3 | 33.8 | **38.7** | 43.9 |
| | F1 | 22.8 | 34.6 | 29.0 | 42.8 | 31.0 | 42.9 | 36.8 | 39.3 | 40.2 | **45.6** | - |

Table 1: Results table for the experiments on SearchQA. Two-sample *t*-tests between the AQA results and either the Base-NMT or the MI-SubQuery results show that differences in F1 and Exact Match scores are statistically significant, $p < 10^{-4}$, for both Top Hypothesis and CNN predictions. The difference between Base-NMT and MI-SubQuery is also significant for Top Hypothesis predictions.

for a heuristic weighted voting scheme to implement deterministic selection. Let $a$ be the answer returned by BiDAF for query $q$, with an associated score $s(a)$. We pick the answer according to $\text{argmax}_a \sum_{a'=a} s(a')$. In *AQA MaxConf* we select the answer with the single highest BiDAF score across question reformulations. Finally, *AQA CNN* identifies the complete system with the learned CNN model described in Section 3.

Table 1 shows the results. We report exact match (EM) and F1 metrics, computed on token level between the predicted answer and the gold answer. We present results on the full validation and test sets (referred to as *n-gram* in (Dunn et al., 2017)). Overall, SearchQA appears to be harder than other recent QA tasks such as SQuAD (Rajpurkar et al., 2016), for both machines and humans. BiDAF's performance drops by 40 F1 points on SearchQA compared to SQuAD. However, BiDAF is still competitive on SeachQA, improving over the Attention Sum Reader network by 13.7 F1 points.

Using the top hypothesis already yields an improvement of 2.2 F1 on the test set. This demonstrates that even the reformulator alone is capable to produce questions more easily answered by the environment. When generating a single prediction, both MI-SubQuery and Base-NMT benchmarks perform worse than BiDAF. Heuristic selection via both Voting and Max Conf yields a further performance boost. Both heuristics draw upon the intuition that when BiDAF is confident in its answer it is more likely to be correct, and that multiple instances of the same answer provide positive evidence (for MaxConf, the max operation implicitly rewards having an answer scored with respect to multiple questions). Finally, a trained selection function improves performance further, yielding an absolute increase of 11.4 F1 points (32% relative) over BiDAF with the original questions. In terms of exact match score, this more than closes half the gap between BiDAF and human performance. The benchmarks improve considerably when they generate $N$ candidates, and paired with a dedicated CNN selector. This is not surprising as it provides an ensemble prediction setup. However, the AQA CNN system outperforms both MI-SubQuery and Base-NMT in all conditions by about 3%.

Finally, we consider the maximum performance possible that could be achieved by picking the answer with the highest F1 score from the set of those returned for all available reformulations. Here we find that the different sources of rewrites provide comparable headroom: the oracle Exact Match is near 50, while the oracle F1 is close to 58.

## 6 ANALYSIS OF THE AGENT'S LANGUAGE

The AQA agent can learn several types of sub-optimal policies. For example, it can converge to deterministic policies by learning to emit the same, meaningless, reformulation for any input question. This strategy can lead to local optima because the environment has built in strong priors on what looks like a likely answer, even ignoring the input question. Hence, convergence to non-negligible performance is easy. Entropy regularization typically fixes this behavior. Too much weight on the entropy regularizer, on the other hand, might yield random policies. A more competitive sub-optimal policy is one that generates minimal changes to the input, in order to stay close to the original question. This is a successful strategy because the environment has been trained on the original questions alone, which leads to baseline performance.

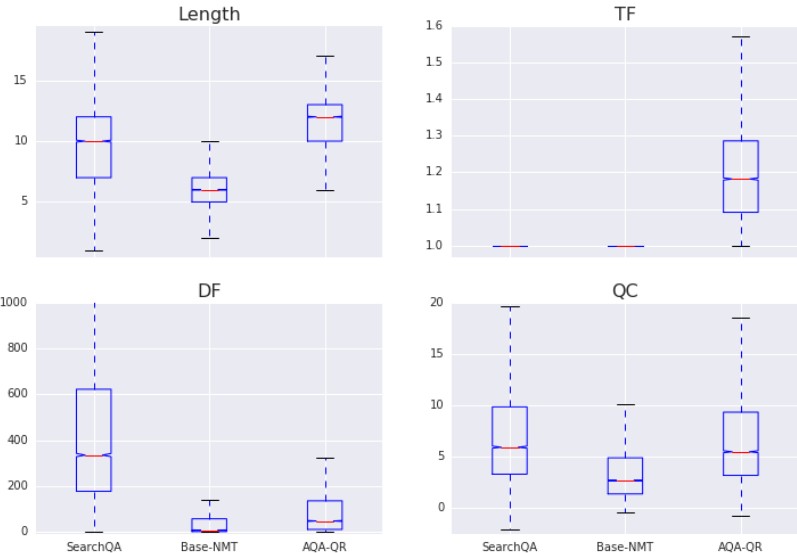

Figure 2: Boxplot summaries of the statistics collected for all types of questions. Two-sample $t$-tests performed on all pairs in each box confirm that the differences in means are significant $p < 10^{-3}$.

It seems quite remarkable then that AQA is able to learn non-trivial reformulation policies, that differ significantly from all of the above. One can think of the policy as a language for formulating questions that the agent has developed while engaging in a machine-machine communication with the environment. In this section we look deeper into the agent's language.

## 6.1 GENERAL PROPERTIES

We analyze input questions and reformulations on the development partition of SearchQA to gain insights on how the agent's language evolves during training via policy gradient. It is important to note that in the SearchQA dataset the original *Jeopardy!* clues have been preprocessed by lower-casing and stop word removal. The resulting preprocessed clues that form the sources (inputs) for the sequence-to-sequence reformulation model resemble more keyword-based search queries than grammatical questions. For example, the clue *Gandhi was deeply influenced by this count who wrote "War and Peace"* is simplified to *gandhi deeply influenced count wrote war peace*.

The (preprocessed) SearchQA questions contain 9.6 words on average. They contain few repeated terms, computed as the mean term frequency (TF) per question. The average is 1.03, but for most of the queries (75%) TF is 1.0. We also compute the median document frequency (DF) per query, where the document is the context from which the answer is selected, as a measure of how informative a term is.[4] As another measure of query performance, we also compute Query Clarity (QC) (Cronen-Townsend et al., 2002).[5] Figure 2 summarizes statistics of the questions and rewrites.

We first consider the top hypothesis generated by the pre-trained NMT reformulation system, before reinforcement learning (Base-NMT). The Base-NMT rewrites differ greatly from their sources. They are shorter, 6.3 words on average, and have even fewer repeated terms (1.01). Interestingly, these reformulations are mostly syntactically well-formed questions. For example, the clue above becomes *Who influenced count wrote war?*.[6] Base-NMT improves structural language quality by properly reinserting dropped function words and wh-phrases. We also verified the increased fluency by using a large language model and found that the Base-NMT rewrites are 50% more likely than the original

---

[4]For every token we compute the number of contexts containing that token. Out of these counts, in each question, we take the median instead of the mean to reduce the influence of frequent outliers, such as commas.

[5]The relative entropy between a query language model and the corresponding collection language model. In our case, the document is the context for each query.

[6]More examples can be found in Appendix A.

questions. While more fluent, the Base-NMT rewrites involve lower DF terms. This is probably due to a domain mismatch between SearchQA and the NMT training corpus. The query clarity of the Base-NMT rewrites is also degraded as a result of the transduction process.

We next consider the top hypothesis generated by the AQA question reformulator (AQA-QR) after the policy gradient training. The AQA-QR rewrites are those whose corresponding answers are evaluated as *AQA TopHyp* in Table 1. These single rewrites alone outperform the original SearchQA queries by 2% on the test set. We analyze the top hypothesis instead of the final output of the full AQA agent to avoid confounding effects from the answer selection step. These rewrites look different from both the Base-NMT and the SearchQA ones. For the example above AQA-QR's top hypothesis is *What is name gandhi gandhi influence wrote peace peace?*. Surprisingly, 99.8% start with the prefix *What is name*. The second most frequent is *What country is* (81 times), followed by *What is is* (70) and *What state* (14). This is puzzling as it occurs in only 9 Base-NMT rewrites, and never in the original SearchQA questions. We speculate it might be related to the fact that virtually all answers involve names, of named entities (Micronesia) or generic concepts (pizza).

AQA-QR's rewrites seem less fluent than both the SearchQA and the Base-MT counterparts. In terms of language model probability, they are less likely than both SearchQA and Base-NMT. However, they have more repeated terms (1.2 average TF), are significantly longer (11.9) than in Base-NMT and contain more informative context terms than SearchQA questions (lower DF). Also, the translation process does not affect query clarity much. Finally, we find that AQA-QR's reformulations contain morphological variants in 12.5% of cases. The number of questions that contain multiple tokens with the same stem doubles from SearchQA to AQA-QR. Singular forms are preferred over plurals. Morphological simplification is useful because it increases the chance that a word variant in the question matches the context.

## 6.2 PARAPHRASING QUALITY

We also investigate the general paraphrasing abilities of our model, focusing on the relation between paraphrasing quality and QA quality. To tease apart the relationship between paraphrasing and reformulation for QA we evaluated 3 variants of the reformulator:

**Base-NMT** This is the model used to initialize RL training of the agent. Trained first on the multilingual U.N. corpus, then on the Paralex corpus, as detailed in Section 5.2.
**Base-NMT-NoParalex** This is the model above trained solely on the multilingual U.N. corpus, without the Paralex monolingual corpus.
**Base-NMT+Quora** This is the same as Base-NMT, additionally trained on the Quora dataset[7] which contains 150k duplicate questions.

Following Prakash et al. (2016), we evaluate all models on the MSCOCO[8] (Lin et al., 2014) validation set (VAL2014). This dataset consists of images with 5 captions each, of which we select a random one as the source and the other four as references. We use beam search, to compute the top hypothesis and report uncased, moses-tokenized BLEU using multeval[9] (Clark et al., 2011). Please note, that the MSCOCO data is only used for evaluation purposes. Examples of all systems can be found in Appendix C.

The Base-NMT model performs at 11.4 BLEU (see Table 1 for the QA eval numbers). In contrast, Base-NMT-NoParalex performs poorly at 5.0 BLEU. Limiting training to the multilingual data alone also degrades QA performance: the scores of the Top Hypothesis are at least 5 points lower in all metrics and CNN scores are 2-3 points lower.

By training on additional monolingual data, the Base-NMT+Quora model improves BLEU score slightly to 11.6. End-to-end QA performance also improves marginally, the maximum delta with respect to Base-NMT under all conditions is +0.5 points, but the difference is not statistically significant. Thus, adding the Quora training does not have a significant effect. This might be due to the fact that most of the improvement is captured by training on the larger Paralex data set. Improving raw paraphrasing quality as well as reformulation fluency helps AQA up to a point. However, they are only partially aligned with the main task, which is QA performance. The AQA-QR reformulator

---

[7] https://data.quora.com/First-Quora-Dataset-Release-Question-Pairs
[8] http://cocodataset.org/
[9] https://github.com/jhclark/multeval

has a BLEU score of 8.6, well below both Base-NMT models trained on monolingual data. Yet, AQA-QR significantly outperforms all others in the QA task. Training the agent starting from the Base-NMT+Quora model yielded comparable results as starting from Base-NMT.

## 6.3 DISCUSSION

Recently, Lewis et al. (2017) trained chatbots that negotiate via language utterances in order to complete a task. They report that the agent's language diverges from human language if there is no incentive for fluency in the reward function. Our findings seem related. The fact that the questions reformulated by AQA do not resemble natural language is not due to the keyword-like SearchQA input questions, because Base-NMT is capable of producing more fluent questions from the same input. AQA learns to re-weight terms by focusing on informative (lower document frequency), query-specific (high query clarity), terms while increasing term frequency (TF) via duplication. At the same time it learns to modify surface forms in ways akin to stemming and morphological analysis.

Some of the techniques seem to adapt to the specific properties of current deep QA architectures such as character-based modeling and attention. Sometimes AQA learns to generate semantically nonsensical, novel, surface term variants; e.g., it might transform the adjective *dense* to *densey*. The only justification for this is that such forms can be still exploited by the character-based BiDAF question encoder. Finally, repetitions can directly increase the chances of alignment in the attention components.

We hypothesize that, while there is no incentive for the model to use human language due to the nature of the task, AQA learns to ask BiDAF questions by optimizing a language that increases the likelihood of BiDAF *ranking* better the candidate answers. Jia & Liang (2017) argue that reading comprehension systems are not capable of significant language understanding and fail easily in adversarial settings. We speculate that current machine comprehension tasks involve mostly pattern matching and relevance modeling. As a consequence deep QA systems might implement sophisticated ranking systems trained to sort snippets of text from the context. As such, they resemble document retrieval systems which incentivizes the (re-)discovery of IR techniques, such as tf-idf re-weighting and stemming, that have been successful for decades (Baeza-Yates & Ribeiro-Neto, 1999).

## 7 CONCLUSION

We propose a new framework to improve question answering. We call it active question answering (AQA), as it aims to improve answering by systematically perturbing input questions. We investigated a first system of this kind that has three components: a question reformulator, a black box QA system, and a candidate answer aggregator. The reformulator and aggregator form a trainable agent that seeks to elicit the best answers from the QA system. Importantly, the agent may only query the environment with natural language questions. Experimental results prove that the approach is highly effective and that the agent is able to learn non-trivial and somewhat interpretable reformulation policies.

For future work, we will continue developing active question answering, investigating the sequential, iterative aspects of information seeking tasks, framed as end-to-end RL problems, thus, closing the loop between the reformulator and the selector.

## 8 ACKNOWLEDGEMENTS

We would like to thank the anonymous reviewers for their valuable comments and suggestions. We would also like to thank Jyrki Alakuijala, Gábor Bártok, Alexey Gronskiy, Rodrigo Nogueira and Hugo Penedones for insightful discussions and technical feedback.

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

# A  REFORMULATION EXAMPLES

Table 2: Results of the qualitative analysis on SearchQA. For the original *Jeopardy!* questions we give the reference answer, otherwise the answer given by BiDAF.

| Model | Query | Reference / Answer from BiDAF (F1) |
|---|---|---|
| Jeopardy! | People of this nation AKA Nippon wrote with a brush, so painting became the preferred form of artistic expression | japan |
| SearchQA | people nation aka nippon wrote brush , painting became preferred form artistic expression | japan (1.0) |
| MI | nippon brush preferred | julian (0) |
| Base-NMT | Aka nippon written form artistic expression? | julian (0) |
| AQA-QR | What is name did people nation aka nippon wrote brush expression? | japan (1.0) |
| AQA-Full | people nation aka nippon wrote brush , painting became preferred form artistic expression | japan (1.0) |
| Jeopardy! | Michael Caine & Steve Martin teamed up as Lawrence & Freddy, a couple of these, the title of a 1988 film | dirty rotten scoundrels |
| SearchQA | michael caine steve martin teamed lawrence freddy , couple , title 1988 film | dirty rotten scoundrels (1.0) |
| MI | caine teamed freddy | dirty rotten scoundrels (1.0) |
| Base-NMT | Who was lawrence of michael caine steve martin? | rain man 1988 best picture fikkle [... 25 tokens] (0.18) |
| AQA-QR | What is name is name is name michael caine steve martin teamed lawrence freddy and title 1988 film? | dirty rotten scoundrels (1.0) |
| AQA-Full | What is name is name where name is name michael caine steve martin teamed lawrence freddy and title 1988 film key 2000 ? | dirty rotten scoundrels (1.0) |
| Jeopardy! | Used underwater, ammonia gelatin is a waterproof type of this explosive | dynamite |
| SearchQA | used underwater , ammonia gelatin waterproof type explosive | nitroglycerin (0) |
| MI | ammonia gelatin waterproof | nitroglycerin (0) |
| Base-NMT | Where is ammonia gelatin waterproof? nitroglycerin (0) | |
| AQA-QR | What is name is used under water with ammonia gelatin water waterproof type explosive? | nitroglycerin (0) |
| AQA-Full | used underwater , ammonia gelatin waterproof type explosive | nitroglycerin (0) |
| Jeopardy! | The Cleveland Peninsula is about 40 miles northwest of Ketchikan in this state | alaska |
| SearchQA | cleveland peninsula 40 miles northwest ketchikan state | alaska 's community information summary says [... 113 tokens] (0.02) |
| MI | cleveland peninsula ketchikan | alaska 's dec 16 , 1997 [... 132 tokens] (0.01) |
| Base-NMT | The cleveland peninsula 40 miles? | ketchikan , alaska located northwest tip [... 46 tokens] (0.04) |
| AQA-QR | What is name is cleveland peninsula state northwest state state state? | alaska (1.0) |
| AQA-Full | What is name are cleveland peninsula state northwest state state state ? | alaska (1.0) |
| Jeopardy! | Tess Ocean, Tinker Bell, Charlotte the Spider | julia roberts |
| SearchQA | tess ocean , tinker bell , charlotte spider | julia roberts tv com charlotte spider [... 87 tokens] (0.04) |
| MI | tess tinker spider | julia roberts tv com charlotte spider [... 119 tokens] (0.01) |
| Base-NMT | What ocean tess tinker bell? | julia roberts american actress producer made [... 206 tokens] (0.02) |
| AQA-QR | What is name tess ocean tinker bell link charlotte spider? | julia roberts (1.0) |
| AQA-Full | What is name is name tess ocean tinker bell spider contain charlotte spider contain hump around the world winter au to finish au de mon moist | julia roberts (1.0) |
| Jeopardy! | During the Tertiary Period, India plowed into Eurasia & this highest mountain range was formed | himalayas |
| SearchQA | tertiary period , india plowed eurasia highest mountain range formed | himalayas (1.0) |
| MI | tertiary plowed eurasia | himalayas (1.0) |
| Base-NMT | What is eurasia highest mountain range? | himalayas (1.0) |
| AQA-QR | What is name were tertiary period in india plowed eurasia? | himalayas (1.0) |
| AQA-Full | tertiary period , india plowed eurasia highest mountain range formed | himalayas (1.0) |
| Jeopardy! | The melody heard here is from the opera about Serse, better known to us as this "X"-rated Persian king | xerxes |
| SearchQA | melody heard opera serse , better known us x rated persian king | gilbert sullivan (0) |
| MI | melody opera persian | gilbert sullivan (0) |
| Base-NMT | Melody heard opera serse thing? | gilbert sullivan (0) |
| AQA-QR | What is name melody heard opera serse is better persian king? | gilbert sullivan (0) |
| AQA-Full | What is name is name melody heard opera serse is better persian king persian K ? | gilbert sullivan (0) |

# B  EXAMPLES OF RANKING LOSSES

Table 3: Examples of queries where none of the methods produce the right answer, but the Oracle model can.

| Model | Query | Answer (F1) |
|---|---|---|
| SearchQA | ancient times , pentelic quarries major source building material athens | parthenon (0.0) |
| AQA-QR | What is name an Ancient History Warry quarry material athens? | parthenon (0.0) |
| AQA-CNN | ancient times , pentelic quarries major source building material athens | parthenon (0.0) |
| AQA-Oracle | What is name is name an Ancient Romes For pentelic quarry material athens measure athens? | marble (1.0) |
| SearchQA | 1949 1999 germany 's bundestag legislature met city | berlin (0.0) |
| AQA-QR | What is name is name 1999 germanyś bundestag legislature met city city? | berlin (0.0) |
| AQA-CNN | What is name is name 1999 germany germany bundestag legislature city city? | berlin (0.0) |
| AQA-Oracle | What is name is name a 1999 germany germany legislature met city city? | bonn (1.0) |
| SearchQA | utah jazz retired 12 jersey | karl malone (0.0) |
| AQA-QR | What is name did utah jazz retired jersey jersey? | karl malone (0.0) |
| AQA-CNN | What is name did utah jazz retired jersey jersey? | karl malone (0.0) |
| AQA-Oracle | What is name is name where utah jazz is retired jersey jersey? | john stockton (1.0) |
| SearchQA | swing intersection count basie capital virginia | chicago (0.0) |
| AQA-QR | What is name for swing intersection count basie capital virginia? | chicago (0.0) |
| AQA-CNN | swing intersection count basie capital virginia | chicago (0.0) |
| AQA-Oracle | What is name swing swing intersection capital virginia? | richmond (1.0) |
| SearchQA | huge upset virginia , jim webb beat senator prospective presidential candidate | hillary clinton (0.0) |
| AQA-QR | What is name is name huge upset virginia jug webb beat senator? | jim webb (0.0) |
| AQA-CNN | What is name is name huge upset virginia jug webb beat senator world candidate? | jim webb (0.0) |
| AQA-Oracle | What is name is name huge upset virginia jim webb beat senator? | george allen (1.0) |

# C    PARAPHRASING EXAMPLES

Table 4: Paraphrasing examples on captions from MSCOCO (Lin et al., 2014).

| | |
|---|---|
| Source | a man riding a wave on top of a surfboard. |
| Reference | a man on a surfboard in the crest of a wave. |
| Base-NMT | a man on top of a surfboard? |
| Base-NMT-NoParalex | a mankind arises away from a floor. |
| Base-NMT+Quora | it possible to riding a wave on a surfboard? |
| AQA-QR | what is name an average man riding a wave at onfboard? |
| Source | a person riding some skis through a snowy mountain trail. |
| Reference | a man on skis is standing by some trees. |
| Base-NMT | what is some skis through a snowy mountain trail? |
| Base-NMT-NoParalex | the person rising somekises through a snow blood. |
| Base-NMT+Quora | what is a skis mountain trail? |
| AQA-QR | what is name an average person riding some one snowy mountain trail? |
| Source | a group of people all taking a bite out of the same sandwich. |
| Reference | a group of people eating rom a large sandwich. |
| Base-NMT | what do people call a bite out of the same sandwich? |
| Base-NMT-NoParalex | a group of people who takes a bargained from the same index. |
| Base-NMT+Quora | what is the group of people who take bite out of the same sandwich? |
| AQA-QR | what is name a group of people with bite out of sandwich? |
| Source | a partially lit asian person holding a phone |
| Reference | a man standing next to a sign talking on a phone. |
| Base-NMT | what is a partical that holds a phone? |
| Base-NMT-NoParalex | plus little as an associate person |
| Base-NMT+Quora | what is a partially little asian person? |
| AQA-QR | what is name is associated with a partially active asian person holding a phone? |
| Source | a car driving through a very flooded street |
| Reference | a car is driving through a flooded street. |
| Base-NMT | what car driving through a flooded street? |
| Base-NMT-NoParalex | analysing through a heat street |
| Base-NMT+Quora | i driving through a flooded street? |
| AQA-QR | what is name is given to a car with a car driving through a flooded street? |
| Source | a man is holding a tennis racket on a court |
| Reference | a man holding a racquet on top of a tennis court. |
| Base-NMT | how do you hold a tennis racket? |
| Base-NMT-NoParalex | a man to hold a threshold on a court |
| Base-NMT+Quora | what is a tennis racket? |
| AQA-QR | what is name an average man make a tennis racket on a court? |
| Source | a pizza on a pan that is half gone |
| Reference | a table with a half eaten pizza and glasses if beer. |
| Base-NMT | what is half gone in a pizza? |
| Base-NMT-NoParalex | average on a size |
| Base-NMT+Quora | is a pizza on a pan that is half gone? |
| AQA-QR | what is name a pizza on a pizza pan that is half gone? |
| Source | a boy holding a glove while on a baseball field. |
| Reference | a young boy bending over holding a catchers mitt. |
| Base-NMT | how do you hold a glove on a baseball field? |
| Base-NMT-NoParalex | a cell holding a globe at a later field. |
| Base-NMT+Quora | is it possible to hold a glove while on a baseball field? |
| AQA-QR | what is name an boy make up a glove while on baseball field? |
| Source | a street light with a red arrow pointing left. |
| Reference | a tagged traffic light displays a red arrow. |
| Base-NMT | a street light with red arrow pointing? |
| Base-NMT-NoParalex | a strength with a skilling point. |
| Base-NMT+Quora | what is the street light of a red arrow pointing? |
| AQA-QR | what is name a street light with red arrow pointing left? |

