# OpenReview forum: "Ask the Right Questions: Active Question Reformulation with Reinforcement Learning"
_ICLR.cc/2018/Conference — Accept (Oral)_

### Official Review · AnonReviewer2 · 2017-11-25
**good, but could be better with more details and experiments**

**Rating:** 7
**Confidence:** 5

**Review:**

This paper proposes active question answering via a reinforcement learning approach that can learn to rephrase the original questions in a way that can provide the best possible answers. Evaluation on the SearchQA dataset shows significant improvement over the state-of-the-art model that uses the original questions.

In general, the paper is well-written (although there are a lot of typos and grammatical errors that need to be corrected), and the main ideas are clear. It would have been useful to provide some more details and carry out additional experiments to strengthen the merit of the proposed model.

Especially, in Section 4.2, more details about the quality of paraphrasing after training with the multilingual, monolingual, and refined models would be helpful. Which evaluation metrics were used to evaluate the quality? Also, more monolingual experiments could have been conducted with state-of-the-art neural paraphrasing models on WikiQA and Quora datasets (e.g. see https://arxiv.org/pdf/1610.03098.pdf and https://arxiv.org/pdf/1709.05074.pdf).

More details with examples should be provided about the variants of AQA along with the oracle model. Especially, step-by-step examples (for all alternative models) from input (original question) to question reformulations to output (answer/candidate answers) would be useful to understand how each module/variation is having an impact towards the best possible answer/ground truth.

Although experiments on SearchQA demonstrate good results, I think it would be also interesting to see the results on additional datasets e.g. MS MARCO (Nguyen et al., 2016), which is very similar to the SearchQA dataset, in order to confirm the generalizability of the proposed approach.

---------------------------------------------
Thanks for revising the paper, I am happy to update my scores.

---

> ### Author Response · Authors · 2017-12-20
> **Response to AnonReviewer2**
>
> Thanks for your review and suggestions! We address each point below:
>
> Other datasets
> We agree that it will be important to extend the empirical evaluation on new datasets. Our current experimental setup cannot be straightforwardly applied to MsMarco, unfortunately. Our environment (the BiDAF QA system) is an extractive QA system. However, MsMarco contains many answers (55%) that are not substrings of the context; even after text normalization, 36% are missing. We plan to investigate the use of generative answer models for the environment with which we could extend AQA to this data.
>
> Paraphrasing quality
> Regarding stand-alone evaluation of the paraphrasing quality of our models, we ran several additional experiments inspired by the suggested work.
> We focused on the relation between paraphrasing quality and QA quality. To tease apart the relationship between paraphrasing and reformulation for QA we evaluated 3 variants of the reformulator:
>
> Base-NMT: this is the model used to initialize RL training of the agent. Trained first on the multilingual U.N. corpus, then on the Paralex corpus.
> Base-NMT-NoParalex: is the model above trained solely on the multilingual U.N. corpus, without the Paralex monolingual corpus.
> Base-NMT+Quora: is the same as Base-NMT, additionally trained on the Quora duplicate question dataset.
>
> Following Prakash et al. (2016) we evaluated all models on MSCOCO, selecting one out of five captions at random from Val2014 and using the other 4 as references. We use beam search, as in the paper, to compute the top hypothesis and report uncased, moses-tokenized BLEU using John Clark's multeval. [github.com/jhclark/multeval]
> The Base-NMT model performs at 11.4 BLEU (see Table 1 for the QA eval numbers). Base-NMT-NoParalex performs poorly at 5.0 BLEU. Limiting training to the multilingual data alone also degrades QA performance: the scores of the Top Hypothesis are at least 5 points lower in all metrics, CNN scores are 2-3 points lower for all metrics.
> By training on additional monolingual data, the Base-NMT+Quora model BLEU score improves marginally to 11.6. End-to-end QA performance also improves marginally, the maximum delta with respect to Base-NMT under all conditions is +0.5 points, but the difference is not statistically significant. Thus, adding the Quora training does not have a significant effect. This might be due to the fact that most of the improvement is captured by training on the larger Paralex data set.
>
> Improving raw paraphrasing quality as well as reformulation fluency help AQA up to a point. However, they are only partially aligned with the main task, which is QA performance. The AQA-QR reformulator has a BLEU score of 8.6, well below both Base-NMT models trained on monolingual data. AQA-QR significantly outperforms all others in the QA task. Training the agent starting from the Base-NMT+Quora model yielded identical results as starting from Base-NMT.
>
> Examples
> We have updated Appendix A in the paper with the answers corresponding to all queries, together with their F1 scores. We also added a few examples (Appendix B) where the agent is not able to identify the correct candidate reformulation, even if present in the candidate set. We also added an appendix (C) with example paraphrases from MSCOCO from the different models.
>
> Presentation
> We spelled and grammar checked the manuscript.

---

### Official Review · AnonReviewer1 · 2017-11-27
**Well written paper that clearly describes a RL-based active approach for question reformulation and answer selection with interesting experimental results.**

**Rating:** 8
**Confidence:** 3

**Review:**

This article clearly describes how they designed and actively trained 2 models for question reformulation and answer selection during question answering episodes. The reformulation component is trained using a policy gradient over a sequence-to-sequence model (original vs. reformulated questions). The model is first pre-trained using a bidirectional LSTM on multilingual pairs of sentences. A small monolingual bitext corpus is the uses to improve the quality of the results. A CNN binary classifier performs answer selection.

The paper is well written and the approach is well described. I was first skeptical by the use of this technique but as the authors mention in their paper, it seems that the sequence-to-sequence translation model generate sequence of words that enables the black box environment to find meaningful answers, even though the questions are not semantically correct. Experimental clearly indicates that training both selection and reformulation components with the proposed active scheme clearly improves the performance of the Q&A system.

---

> ### Author Response · Authors · 2017-12-20
> **Thanks, AnonReviewer1**
>
> We thank Reviewer 1 for the encouraging feedback!

---

### Official Review · AnonReviewer3 · 2017-11-29
**RL formulation for question rewrite, but with weak justification**

**Rating:** 6
**Confidence:** 4

**Review:**

This paper formulates the Jeopardy QA as a query reformulation task that leverages a search engine.  In particular, a user will try a sequence of alternative queries based on the original question in order to find the answer.  The RL formulation essentially tries to mimic this process.  Although this is an interesting formulation, as promoted by some recent work, this paper does not provide compelling reasons why it's a good formulation.  The lack of serious comparisons to baseline methods makes it hard to judge the value of this work.

Detailed comments/questions:
	1. I am actually quite confused on why it's a good RL setting. For a human user, having a series of queries to search for the right answer is a natural process, but it's not natural for a computer program.  For instance, each query can be viewed as different formulation of the same question and can be issued concurrently. Although formulated as an RL problem,  it is not clear to me whether the search result after each episode has been used as the immediate environment feedback. As a result, the dependency between actions seems rather weak.
	2. I also feel that the comparisons to other baselines (not just the variation of the proposed system) are not entirely fair. For instance, the baseline BiDAF model has only one shot, namely using the original question as query.  In this case, AQA should be allowed to use the same budget -- only one query.  Another more realistic baseline is to follow the existing work on query formulation in the IR community.  For example, 20 shorter queries generated by methods like [1] can be used to compare the queries created by AQA.

[1] Kumaran & Carvalho. "Reducing Long Queries Using Query Quality Predictors". SIGIR-09

Pros:
	1. An interesting RL formulation for query reformulation

Cons:
	1. The use of RL is not properly justified
	2. The empirical result is not convincing that the proposed method is indeed advantageous

---------------------------------------

After reading the author response and checking the revised paper, I'm both delighted and surprised that the authors improved the submission substantially and presented stronger results.  I believe the updated version has reached the bar and recommend accepting this paper.

---

> ### Author Response · Authors · 2017-12-20
> **Response to AnonReviewer3**
>
> Thanks for your review, questions, and suggestions which we address below:
>
> 1- RL formulation
> We require RL (policy gradient) because (a) the reward function is non-differentiable, and (b) we are optimizing against a black box environment using only queries, i.e. no supervised query transformation data (query to query that works better for a particular QA system) is available.
> Without RL we could not optimize these reformulations against the black-box environment to maximize expected answer quality (F1 score).
>
> Regarding the training process you are correct: in this work, the reformulations of the initial query are indeed issued concurrently, as shown in Figure 1. We note this when we introduce the agent in the first paragraph of Section 2; we say “The agent then generates a *set* of reformulations {q_i}” rather than a sequence.
>
> In the last line of the conclusion, we comment that we plan to extend AQA to sequential reformulations which would then depend on the previous questions/answers also.
>
> 2- Baseline comparisons
> We computed an IR baseline following [Kumaran & Carvalho, 2009] as suggested. We implemented the candidate generation method (Section 4.3) of their system to generate subquery reformulations of the original query. We choose the reformulations from the term-level subsequences of length 3 to 6. We associate each reformulation with a graph, where the vertices are the terms and the edges are the mutual information between terms. We rank the reformulations by the average edge weights of the Maximum Spanning Trees of the corresponding graphs. We keep the top 20 reformulations, the same number as we keep for the AQA agent. Then, we train a CNN to score these reformulations to identify those with above-average F1, in exactly the same way we do for the AQA agent. As suggested, we then compare this method both in terms of choosing the single top hypothesis (1 shot), and ensemble prediction (choose from 20 queries).
> We additionally compare AQA to the Base-NMT system in the same way. This is the pre-trained monolingual seq2seq model used to initialize the RL training. We evaluate the Base-NMT model's top hypothesis (1 shot) and in ensemble mode.
>
> We find that the AQA agent outperforms all other methods both in 1-shot prediction (top hypothesis) and using CNNs to pick a hypothesis from 20. To verify that the difference in performance is statistically significant we ran a statistical test. The null hypothesis is always rejected (p<0.00001).
> All results are summarized and discussed in the paper.
>
> PS - After reviewing the suggested paper, and related IR literature we took the opportunity to add an IR query quality metric, QueryClarity, to our qualitative analysis at the end of the paper, in the box plot. QueryClarity contributes to our conclusion. showing that the AQA agent learns to transform the initial reformulations (Base-NMT) into ones that have higher QueryClarity, in addition to having better tf-idf and worse fluency.

---

> > ### Author Response · Authors · 2017-12-21
> > **Updated table**
> >
> > We have summarized the comparison of the AQA agent in different modes versus the baselines in Table 1.

---

### Author Response · Authors · 2017-12-20
**Thanks!**

We would like to thank the reviewers for their valuable comments. It took us as a few weeks to reply because we took the time to implement as much as possible of the feedback. We believe this has benefited the paper significantly. We have uploaded a new version of the pdf with the additional work and reply here to the specific comments in greater detail.

---

### Decision · Program_Chairs · 2018-01-29
**ICLR 2018 Conference Acceptance Decision**

**Decision:**

Accept (Oral)

**Comment:**

this submission presents a novel way in which a neural machine reader could be improved. that is, by learning to reformulate a question specifically for the downstream machine reader. all the reviewers found it positive, and so do i.